# Can Dissociation Mediate the Relationship between Emotional Dysregulation and Intelligence? An Empirical Study Involving Adolescents with and without Complex Trauma Histories

**DOI:** 10.3390/ijerph20031729

**Published:** 2023-01-18

**Authors:** Stefania Cristofanelli, Giorgia Baccini, Eleonora Centonze, Alessandra Colombesi, Marina Cariello, Laura Ferro

**Affiliations:** 1Department of Psychology, Faculty of Psychology, University of Valle d’Aosta, 11100 Aosta, Italy; 2TIARÉ, Association for Mental Health, 10125 Turin, Italy; 3Studio Associato RiPsi, 20129 Milan, Italy

**Keywords:** complex trauma, emotional dysregulation, dissociation, intelligence, adolescence

## Abstract

The main aim of the study was to map the psychological functioning of individuals with adverse childhood experiences, with the objective to characterize developmental trajectories. Specifically, we investigated the relations among three of the seven domains of impairment in children and adolescents who had been exposed to complex trauma. To this end, we tested a mediation model with emotional dysregulation as the independent variable, intelligence as the dependent variable, and dissociation as the mediator. The research sample was composed of 64 participants (10–19 years old); 31 adolescents formed the clinical group and 33 the non-clinical group; for the clinical group, we recruited adolescents who were residents of therapeutic communities and had a history of complex trauma. Both groups completed the Difficulties in emotion regulation scale (DERS), Adolescent dissociative experience scale (A-DES), Trauma symptom checklist for children (TSCC), and Wechsler scales. The data analysis yielded significant results for the control group only. We suggest that healthy adolescents adopt a non-pathological dissociative cognitive style in response to increased emotional dysregulation, thus explaining their enhanced cognitive performance.

## 1. Introduction

During the 1990s, Robert Anda and Vincent Felitti conducted a large-scale study known as “ACE” (Adverse Childhood Experiences), which showed that being exposed to different types of abuse and/or household dysfunction during childhood is associated with a higher risk of developing physical or mental illness during adulthood [1]. Adverse childhood experiences are more common than believed [2]: for example, half of the participants in the ACE study reported having been exposed to events of this kind during childhood [1]. In the United States, is estimated that over 3 million children fall victim to abuse and maltreatment every year [3]. In Italy, the prevalence of child sexual abuse in research samples ranges from a minimum of 12.5% (Varese) to a maximum of 34.1% (Lauria). Furthermore, sexual abuse is a widespread phenomenon throughout the world and is likely to be more frequent than official data suggests [4]. Such interpersonal, cumulative, and early traumatic experiences, which are also referred to in the literature as “complex trauma”, are associated with a vast number of consequences for the mind, brain, and body, in both the short and long terms [2]. The literature suggests that the effects of complex trauma on children and adolescents may be organized into seven areas of impairment: attachment, biology, affect regulation, dissociation, behavioral control, cognition, and self-concept [5]. However, the main diagnostic systems currently used (e.g., the Diagnostic and Statistical Manual of Mental Disorders, DSM-5, or the International Classification of Diseases, ICD-11) do not adequately reflect the multitude of psychological, physical, and physiological outcomes of traumatic events in children and adolescents. The absence of an accurate diagnostic framework that captures all of these developmental effects underpins a phenomenon that is often referred to in the literature as a “hidden epidemic”: namely, many children and adolescents experience medical symptoms that become chronic, and they develop a range of mental health issues in adulthood [2,6]. Practitioners are forced to use multiple psychiatric categories, such as depression, attention-deficit/hyperactivity disorder (ADHD), oppositional defiant disorder (ODD), conduct disorder, anxiety disorder, eating disorders, and others that account for a diversity of symptoms. Nevertheless, each diagnosis portrays only a partial and limited facet of the complex set of clinical issues facing these young people [5]. This is not just a theoretical issue, given that unreliable diagnoses are associated with ineffective treatments, which in turn are linked with the perpetuation of trauma in society. However, healing and preventing traumatic injuries is essential to breaking the cycle of violence, because people who have been hurt end up hurting others [2]. In addition, adolescence is a critical phase because of the neural plasticity that characterizes this period [7]. Given the relatively high mutability in adolescents’ brains, trauma can cause permanent adverse modifications; on the other hand, neural plasticity represents a great opportunity for treatment, because it offers a stronger chance of healing and of successfully helping adolescents to express their potential.

### 1.1. Complex Trauma

The term “complex trauma” stands for interpersonal adverse experiences that typically occur during childhood and that can impact seven areas of functioning (attachment, biology, affect regulation, dissociation, behavioral control, cognition, and self-concept). The events in question include maltreatment, emotional, physical, and sexual abuse, witnessing domestic violence, and losing primary caregivers. These experiences are usually prolonged and/or cumulative and characterized by a sense of betrayal [5].

#### 1.1.1. Attachment

Bowlby defines attachment as a biologically determined tendency to bond with other people [8]. During early childhood, the attachment system is essential to survival because it keeps children physically and emotionally close to their primary caregivers. In the context of this first key relationship, the child builds a representation of self, the other, and self in relation to others. In other words, children develop an internal working model in terms of an emotional and cognitive schema that informs their behavior, cognition, and affectivity [8]. There are four patterns of attachment: secure, which is the adaptive pattern, insecure-avoidant, insecure-ambivalent, and disorganized. Over 80% of maltreated children display a pattern of insecure or disorganized attachment [9]. Indeed, when caregivers are excessively distant, unpredictable, violent, or too preoccupied/distressed to adequately respond to the baby’s needs, the child cannot learn to trust others. A lack of epistemic trust usually leads to increased susceptibility to stress, difficulty with emotional regulation, and altered help-seeking strategies (e.g., dependency or social isolation) [5].

#### 1.1.2. Biology

In response to stressful events, the amygdala activates the hypothalamic–pituitary–adrenal axis (HPA), triggering a “fight or flight” response. More specifically, when the amygdala detects an emotionally significant event, it induces the paraventricular nucleus of the hypothalamus to produce corticotropin release factor (CRF), which stimulates the pituitary gland to secrete adrenocorticotropic hormone (ACTH), which in turn prompts the adrenal cortex to release cortisol [10]. In the immediate term, this chain is physiological and adaptive, but if exposure to stress is prolonged and repeated over time, an excessive amount of cortisol is secreted, and it becomes toxic to the brain [11]. Specifically, high levels of this hormone can seriously damage the hippocampus, a brain structure involved in autobiographical memory, and the affect regulation. Furthermore, children exposed to complex trauma have high concentrations of dopamine and noradrenaline in their urine. High levels of norepinephrine can impair the prefrontal cortex, which is the neural correlate of executive function. The main symptoms of such alterations include difficulty in effecting planning and organization, executing working memory tasks, and suppressing inappropriate responses [12].

#### 1.1.3. Affect Regulation

Problems in the areas of both attachment and neurobiology can lead to issues with affect regulation. Traumatized children and adolescents have problems discriminating among different states of arousal and labeling them, both in self and others. Young people who are exposed to maltreatment, neglect, or abuse experience multiple levels of difficulty with expressing their emotions and regulating their internal experiences, which can lead to chronic emotional numbing and recourse to drug or alcohol abuse as a self-soothing strategy. Furthermore, they tend to react to stressful events with an escalation of responses that can rapidly spin out of control [5]. Finally, emotional dysregulation is a major risk factor for both internal and external mental health disorders, as well as for issues with social adjustment [13].

#### 1.1.4. Dissociation

Children and adolescents who have been exposed to complex trauma can display three different kinds of dissociative symptoms: the automatization of behaviors, compartmentalization of traumatic memories and feelings, and detachment from self and emotions [5]. The automatization of behavior implies that the individual acts without engaging in planning or organization. This places traumatized children and adolescents at greater risk of further victimization and learning difficulties. Compartmentalization concerns the relegation of feelings and memories to implicit memory in an attempt to manage the pain that they can provoke [14]. The consequence of this separation is a disconnection between thoughts, emotions, and somatic experiences: because they cannot be integrated, they are kept outside of the subject’s self-awareness. This can foster the emergence of disorders, such as dissociative amnesia and dissociative identity disorder. A further strategy consists of detachment from self (depersonalization), which entails a feeling of being outside the body and observing oneself as though watching a movie, or from external reality (derealization), which involves the feeling that the world is unreal or that time is passing faster or slower than usual [15].

#### 1.1.5. Behavioral Control

Complex trauma in childhood is associated with poor behavioral regulation, including patterns of both a lack of control, such as aggressive conduct or oppositional defiant disorder [9] and too much control, such as excessive compliance with adults’ requests or rigid control over food intake [16]. Such behavioral patterns permit traumatized children and adolescents to avoid painful emotions and to gain an illusionary sense of mastery. Sometimes, these actions become automatic responses in situations that remind children of their traumatic memories [5].

#### 1.1.6. Cognition

Adverse experiences in early childhood, such as sensory deprivation, maltreatment, neglect, domestic abuse, or violence, can impair cognitive development in children and adolescents [5]. For example, 8-year-old children exposed to domestic violence have lower Intelligence Quotient (IQ) scores than do same-age peers [17]. Furthermore, the more severe the traumatic event, the greater the impact on the intelligence quotient. Physical neglect is associated with poorer performance on memory and attention tasks [18]. Abuse can negatively impact creativity and cognitive flexibility [5], and when it leads to post-traumatic stress disorder (PTSD), it can compromise abstract reasoning and executive function [19]. It is still unclear how complex trauma influences cognition, but it is likely that the neurobiological anomalies caused by repeated exposure to adverse experiences interfere with the normal acquisition of abilities by the impacted brain structures or networks.

#### 1.1.7. Self-Concept

Maltreated or abused children represent themselves as helpless and worthless. In abusive or neglectful families, caregivers do not respond adequately to the child’s needs. Due to these experiences, traumatized children are more likely to anticipate being rejected by others, because they feel defective and unlovable [20]. During childhood, adolescence, and adulthood, the main emotions experienced are shame and guilt, against a backdrop of low self-esteem. When something goes wrong, these young people tend to blame themselves and have great difficulty asking for help [5]. Finally, the frequent ongoing deployment of dissociative mechanisms can lead to the development of a non-integrated, discontinuous, and unstable sense of self [7].

### 1.2. Aims of the Study

The main aim of the study was to map the psychological functioning of individuals exposed to complex trauma, studying the relations among the seven known areas of trauma-related impairment.

From a clinical point of view, the research was designed to increase our knowledge about the constellations of symptoms that can be associated with complex trauma and how they are related to each other. Having a model that links the different symptoms (e.g., outbursts of anger, impaired working memory, derealization, …) would help to prevent them from being classified as manifestations of separate disorders. In other words, it would facilitate the formulation of more accurate diagnoses for informing psychotherapy intervention. Indeed, a further clinical implication of the study concerns trauma treatments and the prevention of adverse mental outcomes. For example, knowing exactly how emotional dysregulation leads to a lower intelligence could orient preventive measures. In pursuit of this general goal, we focus on the associations among emotional dysregulation, cognition, and dissociation in adolescents who had experienced traumatic events as compared to adolescents who had not. We tested a mediation model in order to establish whether emotional dysregulation impacts cognition via dissociative mechanisms. Many studies have described the issues caused by traumatic events in each of the seven areas separately, but only a small number had the aim of exploring the causal relations among them. To our knowledge, the present mediation model (Figure 1) has never been assessed before.

Furthermore, our model appears to meet the need that has been documented in the literature [21] to clarify the underlying pattern of causal relations among emotional dysregulation, dissociation, and intelligence. Indeed, most of the existing studies are based on correlation, confirming the associations among these constructs, but not the direction of the causal mechanisms. For example, from a theoretical point of view, dissociation has alternatively been seen as an emotion regulation strategy or a consequence of failed emotion modulation [21].

### 1.3. Mediation Model: Theoretical Bases

We took complex trauma as the theoretical framework within which to investigate the relations among three of the seven key dimensions of trauma-related impairment: emotional dysregulation, dissociation, and intelligence.

Emotional dysregulation and intelligence. We know from the literature that emotional dysregulation leads to poorer performance in several cognitive tasks and processes, such as attention, working [22] and verbal memory [23], logical and analytical reasoning [24] and decision-making [25]. Difficulty with modulating ‘affect’ can interfere with cognitive processes in a variety of ways, including disrupting the processing of initial information inputs [26], and conditioning the subject’s choice of regulation strategy. For example, if a memory task provokes excessive anxiety, it is likely that the individual will try to suppress this uncomfortable feeling. However, suppressing an emotion absorbs considerable internal resources, which consequently become unavailable for the cognitive task. A more effective strategy would be to reappraise the situation [23].

Emotional dysregulation and dissociation. Dissociation may be viewed as an affect regulation style. It can be seen as a means of suppressing negative emotions [27] or avoiding overwhelming feelings [28]. A similar concept was earlier theorized by Sigmund Freud, who proposed that the displacement mechanism is an unconscious strategy for modifying the target of unacceptable emotions [29]. More recently, displacement has been viewed as a means of coping with stress [30] or transforming aggressiveness into creativity [31]. On the other hand, many authors see dissociation as the outcome of failed attempts to modulate intense emotions [32,33].

Dissociation and intelligence. We also know that high levels of dissociation are associated with poorer performance in a variety of cognitive tasks and competencies, including attention [34,35], working memory [36,37], executive function [38], inhibition of automatic response [39], processing speed [40], and visuospatial functioning [35,41]. Dissociation has also been found to play a role in cognitive disorganization [42].

The research findings offer support for the plausibility of our mediation model. Based on the existing literature, we expected that high levels of emotional dysregulation would be associated with low scores for cognitive dimensions (intelligence quotient, QI, verbal comprehension index, VCI, visual–spatial index, VSI, working memory index, WMI, processing speed index, PSI), and that this relationship would be mediated by dissociation, which would increase as a function of emotional dysregulation. We expected that these effects would be stronger in the clinical group than in the non-clinical controls. The moderated mediation model is presented in Figure 2.

## 2. Materials and Methods

### 2.1. Goals

The purpose of the study was to explore the pattern of relations among emotional dysregulation, dissociation, and cognition in adolescents with a past of complex trauma. To this end, we tested a mediation model with emotional dysregulation as the independent variable, intelligence as the dependent variable, and dissociation as the mediator. The mediated effect was further assessed via a moderated mediation model with “group” as the moderator variable (Figure 2). In other words, we wished to verify whether the relationship between emotional dysregulation and cognition and its mediation by dissociation (varied between the clinical and control group).

### 2.2. Sample

The research data was obtained from a sample of 64 adolescents, divided into 2 groups. The control group comprised 33 participants (M = 14.24 years; SD = 2.09), of whom 19 were female; the clinical group comprised 31 participants (M = 14.14, SD = 2.32), of whom 10 were female. As specified above, the two groups differed in terms of gender distribution. This difference, which is illustrated in Figure 3, is due to recruitment challenges: the clinical group was limited to adolescents resident in therapeutic communities affiliated to the association Tiaré (Bussola, Passeggiata, CER-Il Mago di Oz, Cascina Gasera, Liberi Tutti); furthermore, the control group was recruited using a snowball sampling method, which is practical but not the most targeted way to form a sample

### 2.3. Instruments

Difficulties in emotion regulation scale (DERS). The DERS is a self-report questionnaire comprising 36 items to be rated on a 5-point Likert scale (from 1 = almost never to 5 = almost always), which measures patterns of emotional dysregulation. The test comprises a global scale and six sub-scales assessing non-acceptance, goals, impulse control, awareness, strategies, and clarity, respectively. In this study, participants completed the Italian version of DERS validated by Sighinolfi, Pala, Chiri, MarchettI, and Sica 2010 [43] and Giromini and colleagues 2017 [44]. No Italian validation studies of the DERS have been conducted with adolescents specifically. However, the scale has already been adopted in numerous Italian studies to evaluate emotional dysregulation in adolescents [45,46,47].

Adolescent dissociative experience scale (A-DES). This is a self-report instrument with 30 items to be rated on an 11-point Likert scale (from 0 = “never” to 10 = “always”) that offers a global measure of dissociation. The scale displayed excellent reliability in our sample (α = 0.93), with item-total scale correlation coefficients ranging from 0.20 to 0.74.

Trauma symptom checklist for children (TSCC). The TSCC is a self-report instrument that comprises 54 items to be rated on a 4-point Likert-like scale (from 0 = “never” to 3 = “almost always”). It assesses different kinds of post-traumatic symptoms. The questionnaire is composed of six scales, but we only used the Dissociation scale in this study. It displayed good reliability (number of items = 10, α = 0.78), with item-scale correlation coefficients ranging from 0.23 to 0.63.

Wechsler intelligence scale for children—fourth edition (WISC-IV) and Wechsler adult intelligence scale—fourth edition (WAIS-IV). WISC-IV and WAIS-IV are two psychological tests used to evaluate cognitive abilities in children, adolescents, and adults across five domains of cognitive functioning, and yield the following measures: intelligence quotient (IQ), verbal comprehension index (CVI), visual–spatial index (VSI), working memory index (WMI), and processing speed index (PSI).

### 2.4. Procedure

Participants were recruited using a snowball sampling technique. The inclusion criteria for the control group were: being aged between 10 and 19 years old, never having been diagnosed with a psychological or psychiatric disorder, and not undergoing clinical or pharmacological treatment. The inclusion criteria for the clinical group were: having a history of complex trauma and currently taking part in a residential treatment program at one of the therapeutic communities affiliated with the Tiaré Association. Since only a limited number of adolescents could be recruited from these communities, the sample size was unfortunately very small.

The control group participants individually completed the entire battery of research instruments in a single session. The instruments were administered by trainee practitioners who had received ad hoc training for the task and they were presented in the following order: DERS, A-DES, TSCC, and, finally, WISC-IV or WAIS-IV. The administration took place at the headquarters of the association Tiaré in Turin, 44 Via Claudio Luigi Berthollet. It took a mean of 2.5 h for an adolescent to complete the battery. The clinical group participants completed the self-report instruments in a single session. These instruments were administered by trainee practitioners who had received ad hoc training for the task, and they were presented in the following order: DERS, A-DES, and TSCC. They were administered directly at the therapeutic community where the adolescents were residents and took a mean of one hour to complete. WAIS-IV and WISC-IV were administered separately to the clinical group participants by clinical practitioners as part of a routine psychological assessment.

### 2.5. Statistical Analysis

The data were analyzed using IBM SPSS Statistics software, version 27 (SPSS Inc., Chicago, IL, USA). First, we checked for missing data and verified whether the assumptions of the general linear model (homoscedasticity and normality of the residuals) were met. We then conducted a one-way analysis of variance (ANOVA) in order to compare clinical and control groups in relation to the scores obtained for all Wechsler scale indices and all variables assessed by the self-report instruments.

## 3. Results

Before proceeding with the statistical analysis, we checked for missing data and verified whether the assumptions of the general linear model (homoscedasticity and normality of the residuals) were met.

Missing data. Two subjects in the clinical sample lacked an intelligence quotient score because they obtained a weighted score that was too low to be converted into intelligence quotient points (IQ < 40).

Homoscedasticity. To verify that the residuals of the scores of each dependent variable were distributed in the same way, scatter plots were created with the residuals on the *Y* axis and the predicted values on the *X* axis. In the context of ANOVA, this means testing whether the dependent variable displays similar variance within the groups defined by the independent variable. Observations of the graphs showed that the residuals were arranged along a band of constant amplitude, meaning they were similarly distributed for each of the two groups.

Residue normality distribution. To verify whether the residuals displayed Gaussian distribution, a histogram plot was produced for the residual scores of each variable and the Kolmogorov–Smirnov inferential test was used to test the null hypothesis that the observed residual distribution was normal. All distributions were found to be normal (KS_IQ_ = 0.10, KS_VCI_ = 0.10, KS_VSI_ = 0.06, KS_WMI_ = 0.11, KS_PSI_ = 0.07, KS_DERS_ = 0.08, KS_NONACC_ = 0.16, KS_GOALS_ = 0.07, KS_IMP_ = 0.18, KS_AWAR_ = 0.08, KS_STRAT_ = 0.17, KS_CLAR_ = 0.08, KS_A-DES_ = 0.14, and KS_TSCC-DIS_ = 0.13).

Analysis of variance (ANOVA). In order to verify whether the clinical and control groups differed in relation to any of the constructs under study, the means of the two groups were compared via a series of one-way analyses of variance (ANOVAs). Each ANOVA took the group as the independent variable (IV), and as the dependent variable (DV), the scores for either the global scales or sub-scales of the self-report questionnaires or the indices of the Wechsler scales. The control group obtained significantly higher mean scores than did the clinical group on each of the five cognitive indices, intelligence quotient, IQ, verbal comprehension index, VCI, visual-spatial index, VSI, working memory index, WMI, and processing speed index, PSI (Table 1). In relation to the other variables under study, no significant differences were identified (Table 2).

Moderated mediation: comparison between control and clinical groups.

We assessed multiple moderated mediation models representing possible combinations of the independent variable (IV), dependent variable (DV), and mediator. The analysis showed that the mediation model was only statistically significant for the control group and only for some patterns, which were:Model 1. The first significant model (*p* value = 0.04) featured the total DERS scores as the independent variable (IV), the dissociation scale of TSCC (TSCC_Dissociation) as the mediator, and the processing speed index (PSI) as the dependent variable (DV) (Figure 4);Model 2. The second model that almost reached the threshold for significance (*p* value = 0.06) adopted the non-acceptance sub-scale of DERS as the independent variable (IV), TSCC_Dissociation as the mediator, and the processing speed index (PSI) as the dependent variable (DV) (Figure 5);Model 3. The third significant model (*p* value = 0.03) took the non-acceptance DERS sub-scales as independent variables (IV), TSCC_Dissociation as the mediator, and the PSI as the dependent variable (DV) (Figure 6);Model 4. The fourth model, which reaches the significance threshold (*p* value = 0.05), adopted the awareness sub-scale of DERS as the independent variable (IV), the A-DES scale as the mediator, and PSI as the dependent variable (DV) (Figure 7);Model 5. The fifth model was close to the significance threshold (*p* value = 0.06) and featured the clarity sub-scale of DERS as the independent variable (IV), A-DES as the mediator, and PSI as the dependent variable (DV) (Figure 8);Model 6. The sixth significant model (*p* value = 0.05) featured the clarity sub-scale of DERS as an independent variable (IV), A-DES as the mediator, and the intelligence quotient (IQ) as the dependent variable (DV) (Figure 9).

Table 3 shows the mediated effects and the individual components of the significant mediation models.

### Moderated Mediation Model Controlled for Gender

Since clinical and control groups differ in their gender compositions, in order to control for the possible influence of gender, we repeated the analyses using both gender and group as moderator variables, the total DERS scores as the independent variable (IV), the dissociation scale of A-DES (ADES_Total Score) as the mediator, and the intelligence quotient (IQ) as the dependent variable (DV). Figure 10 represents the new estimated model.

This set of analyses yielded no statistically significant outcomes.

## 4. Discussion

### 4.1. Analysis of Variance (ANOVA)

As reported above, the control group obtained, on average, higher scores than the clinical group for the cognitive variables intelligence quotient (IQ), verbal comprehension index (VCI), visual–spatial index (VSI), working memory index (WMI), and processing speed index (PSI) (Table 1). These outcomes could be due to the years of schooling on cognitive performance, the lack of care and cognitive stimulation, or both. Indeed, the control group participants had completed more years of schooling than the adolescents in the clinical sample; furthermore, the subjects in the clinical group had histories of abuse and neglect, which, as widely documented in the literature, has a negative impact on a range of cognitive abilities. The absence of disorders related to psychological trauma in the control group and the presence of such disorders in the clinical group could account for the significant difference between the mean scores of the two groups on the various cognitive indices. However, we did not measure these variables as part of our research design and it is therefore not possible to reach any empirical conclusions about their influence on cognitive performance. With regard to the scores obtained for the DERS, A-DES, and TSCC (dissociation scale only) self-report questionnaires, the analysis did not point to any significant differences between the mean scores of the two groups (Table 2). Due to a lack of insight into psychological problems and social desirability bias, and despite the complex trauma diagnosis, the genuine absence of dissociation and emotional dysregulation issues may have led to the similarity between the scores of the clinical group and those of the control group.

### 4.2. Moderated Mediation Model

Model 1. In the control group only, as emotional dysregulation increased, levels of dissociation also increased, with a consequent rise in the processing speed. As emotional dysregulation increases a subject’s level of dissociation, this confirms part of our research hypothesis and is in line with existing studies. Furthermore, it should be recalled that adolescence is a period characterized by increased emotional dysregulation and completely natural dissociation and, therefore, it seems plausible that healthy subjects might respond to intense emotions by drawing more frequently on (non-pathological) dissociative mechanisms. Emotional dysregulation appears to be a predictor of processing speed and, in light of the estimated model, this may occur because dissociation intervenes to mediate the relationship. Specifically, dissociation enhances processing speed (PSI). It is possible that when dissociation takes the form of a cognitive style, rather than a psychopathological trait, it may serve to improve attention, working memory, and episodic memory [48]. Thus, conceivably, a higher processing speed may be the outcome of a positive influence of dissociation, as can be the case with other cognitive abilities. In future studies, this hypothesis should be investigated more thoroughly. Another possible explanation is that dissociative tendencies can be associated with a style of information processing in which attention is focused on a limited area of the stimulus. Although such hypotheses have been advanced in relation to patients with dissociative identity disorder [49], we suggest that this kind of attentional style could explain the higher scores of the control group in the present study on the processing speed index (PSI): specifically, the more the participants manage to focus their attention on the task, the faster they may complete it correctly. However, attention is a competence that comes into play across a wide range of cognitive functions, so such an attentional style should have influenced participants’ performances on all indices and not just their processing speed.

Model 2. In the control group only, the tendency to react to challenges with non-acceptance or to experience negative secondary emotions in response to negative primary emotions (also a form of non-acceptance) was associated with higher levels of dissociation as measured via the TSCC. Thus, emotional non-acceptance can lead to increased levels of dissociation. However, such a relationship has previously been documented in patients with histories of trauma [50], while in the present study, it was identified in the control group only. Emotional non-acceptance, expressed in terms of emotional avoidance, appears to be an adaptive mechanism under some circumstances because it decreases the level of distress experienced by the adolescent [50,51]. It should be investigated whether emotional non-acceptance can lead to the development of a non-pathological dissociation between cognition and emotion. If so, this mechanism might ensure that the distress experienced during cognitive tasks does not interfere with the speed of information processing, thus enhancing cognitive performance.

Model 3. In the control group only, more limited access to emotion regulation strategies was related to higher levels of dissociation, as measured using the TSCC. A relationship between emotion regulation strategies and dissociation has been documented in the literature: in general, poor use of coping strategies increases stress [52], while high levels of stress can prompt the deployment of dissociative mechanisms [2]. In future studies, we should further assess whether a narrow repertoire of emotion regulation strategies can prevent stress from being adequately regulated, leading to the implementation of dissociative mechanisms. If such dissociative mechanisms amount to a dissociative cognitive style and not a pathological trait, this might plausibly contribute to superior performance on tasks that demand a high speed of processing.

Model 4. In the control group only, the inability to clearly identify one’s emotions (clarity) was associated with higher levels of dissociation as measured via A-DES. The literature documents a strong relationship between difficulty in identifying and describing feelings, on the one hand, and dissociation on the other, in both clinical [53] and non-clinical [54] subjects. Given that this model was statistically significant, we might propose a lack of emotional clarity as an underlying cause of dissociation, leading in turn to the enhanced speed of information processing. Again, this could potentially constitute a non-pathological dissociative cognitive style, a hypothesis that should be verified in follow-up studies by comparing individuals with pathological and with non-pathological dissociation.

Model 5. In the control group only, not paying attention to emotions and the consequent lack of awareness of emotional reactions was related to higher levels of dissociation as measured via the A-DES. Some authors have proposed an association between low emotional awareness and the use of dissociative mechanisms, without however specifying the causal direction of this relationship [55]. The significance of the present model lends support to the notion of a linear relationship between the two constructs, suggesting that levels of dissociation increase as functions of inattention to one’s own emotional states. As previously hypothesized, if the dissociation takes the form of a non-pathological dissociative style, it could lead to an increase in processing speed.

Model 6. In the control group only, the inability to clearly identify one’s emotions (clarity) was not only associated with faster processing, but also with a higher intelligence quotient (IQ). While the literature supports the existence of a relationship between emotional clarity and dissociation, to our knowledge no studies have investigated the effects of these mechanisms on the intelligence quotient. Generally speaking, it might be hypothesized that dissociation can influence the relationship between emotional dysregulation and cognition, decreasing levels of distress and thus fostering enhanced cognitive performance. However, it is not clear why this pattern of influence was found for the clarity subscale in particular. Similarly, the hypothesis would be more convincing if clarity had had a significant relationship with all the indices in the Wechsler scales.

Overall, at the explanatory level, two points remain unresolved: (1) why were no other patterns of relations found to be statistically significant? (2) why were no significant patterns of relations at all identified in the clinical group? Due to the phenomenon known as publication bias, it is even more difficult to discern why we obtained such a set of results. It is possible that other researchers found similar results to ours but that the statistically insignificant outcomes discouraged them from presenting their work. Although the literature suggests that individuals with a history of trauma can have difficulty regulating their emotions and that this difficulty has a negative effect on intelligence due to the mediation of dissociation, our data do not support this hypothesis. However, one possible explanation for the non-significance of the hypothesized mediation model in the present study is that this model may not offer an adequate description of the relationship between the variables considered. For example, the relations among these three constructs might be non-linear.

### 4.3. Further Considerations about the Sample

Given that there was a significant difference in the gender composition of the two groups (57% of participants in the control group were female as opposed to 32% in the clinical group), we hypothesized that these differences might explain the results obtained in our models. Therefore, we estimated further mediation models using “gender” as a new moderator variable. The aim was to again test the moderated mediation effect, while controlling for the effect of gender. Given that these analyses yielded no significant outcomes, we concluded that gender did not influence the relations among the other variables.

## 5. Conclusions

The limitations of the present study concern both the sample and methodology. First of all, the sample sizes in both the clinical and control groups are barely acceptable for generalizing from the results. Indeed, a small number of participants may fail to yield evidence of relations that, on the contrary, could be significant in larger populations. Furthermore, our analysis suggested that the participants in the clinical and the control groups did not differ on measures of emotional dysregulation and dissociation.

Thus, the low variability in scores for most of the study variables may have masked significant patterns of relationships. To fully explore this hypothesis, a different approach should be used when recruiting clinical and non-clinical samples in future studies. Specifically, the selection process should identify subjects with histories of complex trauma who actually display marked dissociation and emotional dysregulation.

Despite the non-significant results obtained with respect to the clinical group, we may still offer some final considerations that could be of value to mental health practitioners. First, psychotherapy treatments designed to reduce dissociative symptoms and emotional dysregulation could help to prevent declines in cognitive functioning. In other terms, as we showed for the control group, cognitive absorption in tasks enhances cognitive performance. Working with patients to achieve non-pathological levels of dissociation that actually enhance intelligence could be one of the aims of psychotherapy. Second, this study focused on complex trauma to shed light on the highly varied symptomatology that is associated with a history of trauma and to make the scientific community more aware of this diagnosis, which has not been included in any psychopathology manual. The main aim of the study, which was to characterize the potential developmental trajectories of traumatized individuals, may be pursued further in future studies that should investigate the relations among the other areas compromised by traumatic experiences. Another potential extension of this work could be to investigate the role of resilience factors in responding to traumatic experiences, to investigate how resources and protective factors interact with the different areas of mental functioning.

## Figures and Tables

**Figure 1 ijerph-20-01729-f001:**
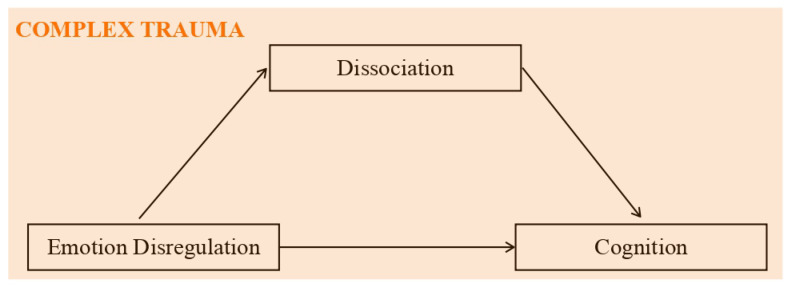
Mediation model: independent variable = emotional dysregulation; dependent variable = cognition; mediator = dissociation.

**Figure 2 ijerph-20-01729-f002:**
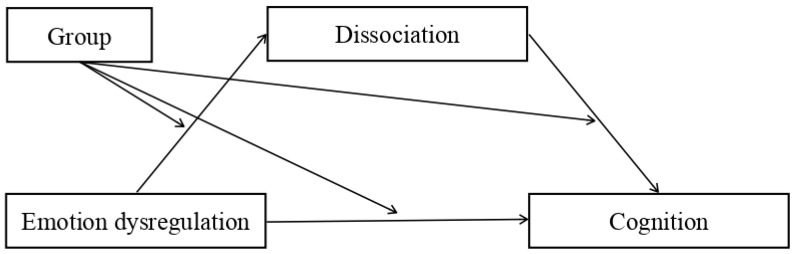
Moderated mediation model: independent variable = emotional dysregulation; dependent variable = cognition; mediator = dissociation; moderator = group.

**Figure 3 ijerph-20-01729-f003:**
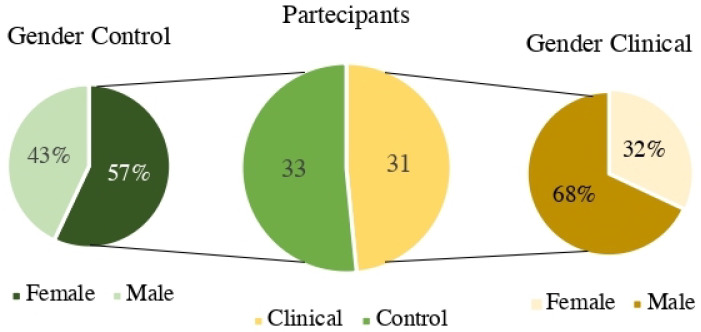
Sample composition: difference in gender distribution between the clinical and control groups.

**Figure 4 ijerph-20-01729-f004:**
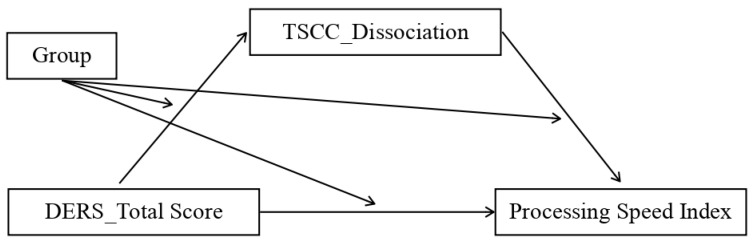
Model 1: IV = total scale of DERS; DV = processing speed index (PSI); mediator = dissociation scale of TSCC.

**Figure 5 ijerph-20-01729-f005:**
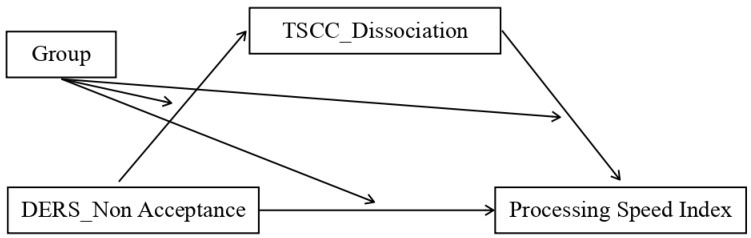
Model 2. IV = non-acceptance scale of DERS; DV = processing speed index (PSI); mediator = dissociation scale of TSCC.

**Figure 6 ijerph-20-01729-f006:**
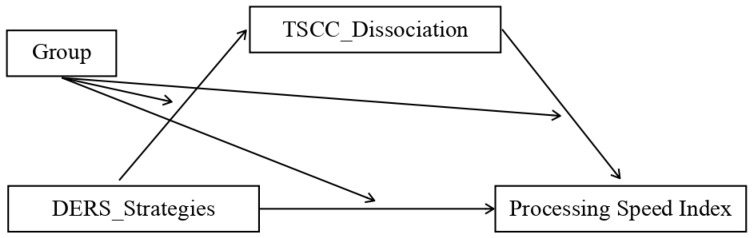
Model 3. IV = non-acceptance scale of DERS; DV = processing speed index (PSI); mediator = dissociation scale of TSCC.

**Figure 7 ijerph-20-01729-f007:**
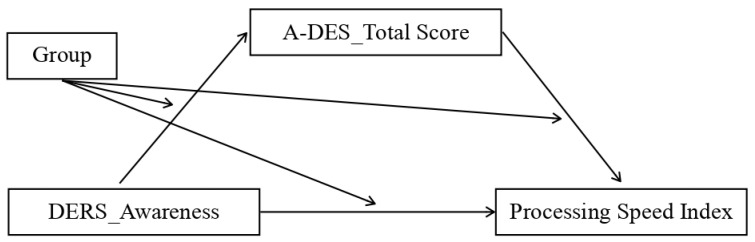
IV = Awareness scale of DERS; DV = processing speed index (PSI); mediator = total scale of A-DES.

**Figure 8 ijerph-20-01729-f008:**
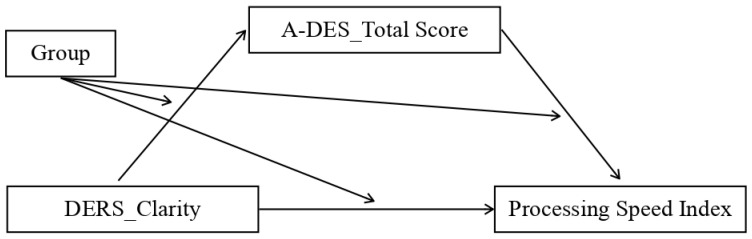
IV = Clarity scale of DERS; DV = processing speed index (PSI); mediator = total scale of A-DES.

**Figure 9 ijerph-20-01729-f009:**
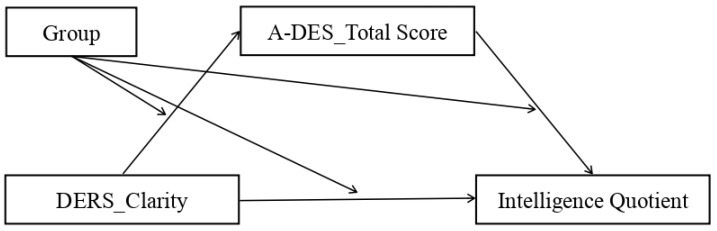
IV = clarity scale of DERS; VD = intelligence quotient (IQ); mediator = total scale of A-DES.

**Figure 10 ijerph-20-01729-f010:**
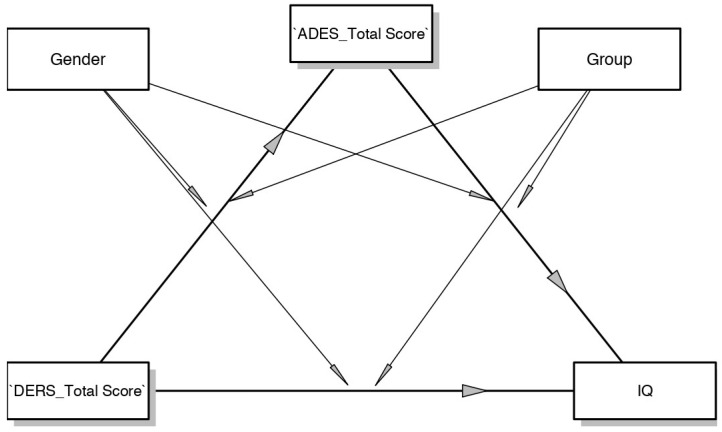
Mediation model moderated by group and gender.

**Table 1 ijerph-20-01729-t001:** Analysis of variance (ANOVA): statistically significant results. IV = independant variable; DV = dependant variable; $R^{2}$ = correlation coefficient; df = degrees of freedom; F = F test; *p* = *p* value; N = number of sample; M = mean; SD = standard deviation; IQ = intelligence quotient; VCI = verbal comprehension index; VSI = visual-spatial index; WMI = working memory index; PSI = processing speed index.

IV = Group					Control Group			Clinical Group		
DV	*R* ^2^	df	F	*p*	N	M	SD	N	M	SD
IQ	0.49	1.60	56.62	<0.001	33	108.67	11.77	29	79.76	18.17
VCI	0.51	1.62	64.08	<0.001	33	110.24	13.28	31	81.39	15.53
VSI	0.30	1.62	26.08	<0.001	33	111.33	13.55	31	87.97	22.26
WMI	0.20	1.62	15.76	<0.001	33	98.33	11.69	31	82.13	20.12
PSI	0.47	1.62	54.02	<0.001	33	103.09	11.94	31	76.97	16.29

**Table 2 ijerph-20-01729-t002:** ANOVAs: non-statistically significant results for all Difficulties in emotion regulation scale (DERS), Adolescent dissociative experience scale (A-DES) and the Trauma symptom checklist for children dissociation scale (TSCC_Dissociation).

IV = Group			
**DV**	**df**	**F**	* **p** *
DERS_Total Score	1.62	0.47	0.50
non-acceptance	1.62	0.03	0.87
goals	1.62	0.58	0.45
impulse	1.62	0.28	0.60
awareness	1.62	2.83	0.10
strategies	1.62	2.77	0.10
clarity	1.62	2.21	0.14
A-DES	1.62	2.14	0.15
TSCC_Dissociation	1.62	0.98	0.33

**Table 3 ijerph-20-01729-t003:** Statistically significant results of moderated mediation model analyses. DERS_TOT = DERS total score; TSCC_DIS = TSCC dissociation scale; NONACC = DERS non-acceptance sub-scale; STRAT = DERS strategies sub-scale; AWAR = DERS awareness sub-scale.

Moderator Level = Control Group			
**Model**	**Effect**	**Beta**	* **p** *
Model 1	DERS_TOT ⇒ TSCC_DIS ⇒ PSI	0.12	0.06
	DERS_TOT ⇒ TSCC_DIS	0.47	0.02
	TSCC_DIS ⇒ PSI	0.26	0.003
Model 2	NONACC ⇒ TSCC_DIS ⇒ PSI	0.10	0.07
	NONACC ⇒ TSCC_DIS	0.43	0.01
	TSCC_DIS ⇒ PSI	0.23	0.01
Model 3	STRAT ⇒ TSCC_DIS ⇒ PSI	0.10	0.11
	STRAT ⇒ TSCC_DIS	0.40	0.06
	TSCC_DIS ⇒ PSI	0.25	0.005
Model 4	AWAR ⇒ A-DES ⇒PSI	0.10	0.10
	AWAR ⇒ A-DES	0.50	0.01
	A-DES ⇒ PSI	0.20	0.02
Model 5	CLARITY ⇒ A-DES ⇒ PSI	0.12	0.09
	CLARITY ⇒ A-DES	0.63	0.002
	A-DES ⇒ PSI	0.19	0.04
Model 6	CLARITY ⇒ A-DES ⇒ IQ	0.11	0.11
	CLARITY ⇒ A-DES	0.63	0.002
	A-DES ⇒ IQ	0.18	0.06

## Data Availability

Restrictions apply to the availability of these data. Data was obtained from TIARÉ Association for Mental Health and are available on request from the corresponding author with the permission of TIARÉ Association for Mental Health.

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
