# Peer review of "Can Dissociation Mediate the Relationship between Emotional Dysregulation and Intelligence? An Empirical Study Involving Adolescents with and without Complex Trauma Histories"

_ijerph, 2023, doi:10.3390/ijerph20031729_

Round 1

Reviewer 1 Report

The article has a very interesting approach and a suitable design. Modifications should be made as soon as the justification of the sample. Firstly, it is a very small sample and secondly there are significant differences that must be explained, especially in its constitution. The control group is 57% female and the clinical group 32%. This methodological aspect should be analyzed, since it can have an impact on the results of the research.

The procedure requires an explanation of the conditions under which the questionnaires were administered. In which place they were completed, which were the professionals who administered them or the time they had to complete them are aspects that are not described in the study.

There are spelling errors in the document, such as a comma instead of a dot at the end of line 224. Dots appear on lines 283 and 302. all acronyms of the text should be developed, for example PTSD.

I hope that the suggestions will support in the progress of the investigation, a cordial greeting.

Author Response

The English revision has been made for the full article. 

Reviewer 2 Report

The mechanism of dissociation between emotional dysregulation and cognitive processes has been known for quite some time. A similar term could be used as 'displacement', which has been described as far back as Sigmund Freud. The displacement effect of emotional experience has also been described in more recent times (McWilliams Nancy., 1990-2000). A variants of dissociation between mental health conditions and psychotic experiences have also been described in contemporary literature (doi: 10.1016/j.jpsychires.2020.08.023).

The mechanism of compensatory increase in information processing speed and higher intelligence in healthy people with emotional disturbances and displacement (dissociation) of childhood emotional problems is also known. The increase in information processing speed is explained by the quite natural compensation of emotional dysregulation by an individual's intellectual efforts.

In this context, the authors need to be clearer about the novelty of the study, because the stated aim (Our goal is to find out if emotion dysregulation, due to complex trauma, can impact on cognition and if this effect could be explained by the intervention of dissociation - Lines 52-53) does not seem new or original. It is possible that the moderated mediation models presented (Lines 301-386) may be stages of social adjustment (and later dysadaptation) of emotionally disturbed people. These people, who have suffered psychological trauma during childhood, have relatively high intelligence in the initial stages, but without appropriate help and psychotherapy will have a natural decline in intelligence. On the other hand, the models presented can help to guide individual psychotherapy for emotional and cognitive disorders in adolescents with complex trauma histories.

The research methodology is in general in line with the aims and objectives of the study.

The conclusions are sufficiently author's vision of the presented results of the study.

It is recommended that the authors supplement the Introduction and Discussion section with a description of the concept of "displacement" with reference to the literature and to specify where the author's idea of "Dissociation" is novel (original).

Author Response

In addition to the revisions suggested, we modified the abstract, the figure 1 showing the general moderated model and figures with each significant model beacause of spelling errors. 

Round 2

Reviewer 1 Report

The paper presents an adequate structure, theorical fundamentacion anda metodology. It is interesting at scientific level.